# LARESA: LOSS-BASED LATENT RESTRUCTURING WITH SEMANTIC ALIGNMENT

## ABSTRACT

Existing solutions for improving the semantic coherence of latent representations from deep classification models often require architectural modifications or pre-training stages. We introduce LARESA, a dynamic, multi-objective, light-weight and regularization-driven training framework that injects semantic priors via auxiliary loss terms into the latent space using only class label text, requiring no architectural changes. LARESA leverages relational distances between language embeddings of the class labels or descriptions, fostering robust and interpretable latent spaces. Our method jointly optimizes traditional classification with semantic alignment and cluster-oriented regularizers with a learnable loss weighting mechanism to encourage both meaningful and well-separated feature representations. Across our experiments, LARESA delivers substantial accuracy improvements while simultaneously enhancing latent space disentanglement. Notably, language embeddings require only a one-time pre-processing step with minimal overhead, even for high-class scenarios, therefore our regularization term introduces negligible computational cost during training, enabling seamless application to existing classification models.

## 1 INTRODUCTION

Despite impressive performance metrics, deep classifiers often rely on probabilistic training dynamics, which undermines their ability to form semantically coherent latent representations (Zhang et al., 2021; Huang et al., 2022). Yet, class discrimination objectives like cross-entropy lack explicit mechanisms to enforce semantic alignment or structure in the latent space (Wen et al., 2016; Cheng et al., 2020). As a result, even well-performing models often learn semantically meaningless latent manifolds or overfit to dataset-specific biases (Geirhos et al., 2020; Tsipras et al., 2019). Cross-entropy optimizes discriminative boundaries but ignores semantic relationships between classes, leading to suboptimal latent spaces where semantically distinct categories cluster closely (e.g., "tiger" and "tulip") or vice versa (Liu et al., 2021; Papyan et al., 2020). This brittleness arises because semantic structure encoded in class labels is discarded during training. Such a lack of structural coherence not only hinders the learning progression and class-wise reasoning, but also leads to latent spaces that are fragile, opaque, and sensitive to noise (Olah et al., 2018; Alain & Bengio, 2016; Engstrom et al., 2019).

We posit that a central limitation of standard classification pipelines is the lack of explicit mechanisms to enforce semantic coherence in the latent space. An ideal representation would cluster semantically similar classes closely and separate dissimilar ones. Multiple works explored to directly structure the latent space according to human manipulations (Geissler et al., 2025; Liao et al., 2025). This yielded improved generalization; however, it was limited by information loss during dimensionality reduction for visualized human manipulation and scalability challenges with increasing class counts due to cognitive constraints.

Explicit alignment with external semantics typically requires dedicated architectures or training stages, such as cross-modal attention in Contrastive Language–Image Pretraining (CLIP) (Radford et al., 2021) or specialized Transformers for structured outputs (Chen et al., 2023), limiting adaptability to existing architectures. The recent maturity of large language models (LLMs) provides an alternative: LLMs encode rich semantic information that can be leveraged to revisit and enhance existing models developed prior to the LLM era. The nuanced similarities are readily exploitable (Kalyan, 2024; Wang

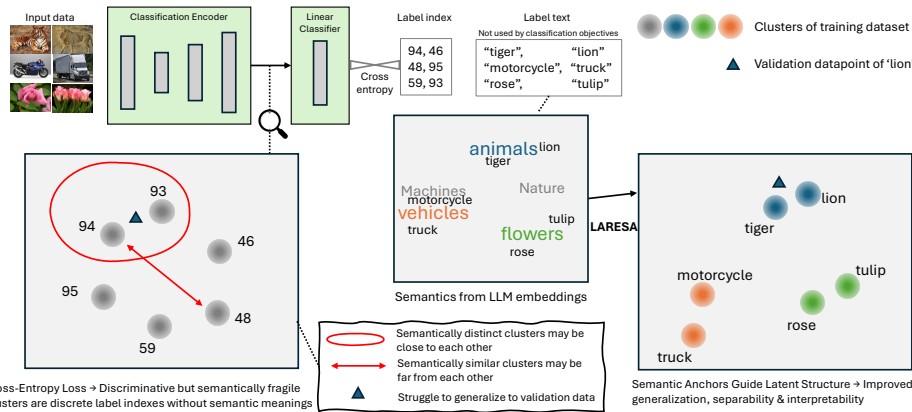

Figure 1: LARESA semantic alignment concept: Cross-entropy training treats class labels as discrete indices, ignoring semantic relationships (e.g., "tiger" "tulip" are only treated as "94" "93"); LARESA leverages existing language embeddings to make use of the label text information.

et al., 2023) through sentence transformers like SBERT(Reimers & Gurevych, 2019a). However, standard classification (especially image) training pipelines relying on loss-based metric learning methods (Barbato et al., 2021; Frascaroli et al., 2024) often lack methods to leverage this prior effectively.

We introduce **LA**tent **RE**structuring with **S**emantic **A**lignment **LARESA**, a loss-based framework that leverages LLM-derived embeddings as semantic priors to guide latent space structuring through auxiliary losses, **bypassing architectural dependencies and pre-training** like CLIP. Our approach enforces alignment through differentiable loss terms alone to enable seamless integration with diverse models and datasets. This mirrors recent trends in flexible regularization (Pati & Lerch, 2021; Frascaroli et al., 2024) but extends them to semantic alignment, avoiding the rigidity of architectural dependencies (Chen et al., 2023). Notably, loss-based regularization also ensures lighter computational overhead (e.g. no additional model components) and more modular adaptation (e.g. bypassing dimension matching) compared to architectural changes.

Our framework employs auxiliary latent losses - cosine-based anchoring loss for alignment of latent clusters with their corresponding semantic embeddings by optimizing for **absolute cluster positioning**; triplet-based loss for semantic consistency that enforces **relational distances** based label dissimilarities; and clustering regularizers promoting **compactness and separability** within class-specific manifolds. These objectives are dynamically weighted together with the cross-entropy loss through learnable weighting coefficients that adaptively rebalance the contributions of semantic and structural objectives during training epochs. We evaluate LARESA across diverse image classification datasets, spanning varying semantic granularity, domain, and backbone architectures. Our experiments demonstrate consistent improvements in classification accuracy, latent structure, and alignment through linguistic embeddings.

In summary, our **contributions** can be summarized:

- We introduce LARESA, a dynamic, loss-based, light-weight and architecture-agnostic training framework that combines semantic alignment from existing LLMs with latent space structuring without the need of expensive pre-training stages.

- We propose cosine-based and triplet-based losses leveraging label embeddings to inject language-derived priors into the learning process, with a learnable loss weighting scheme that automatically balances semantic and discriminative learning objectives during training.

- We empirically validate our method across multiple datasets and architectures, showing that language-guided semantic supervision is a powerful, underutilized source of inductive bias, consistently improving classification accuracy and enhancing latent space disentanglement.

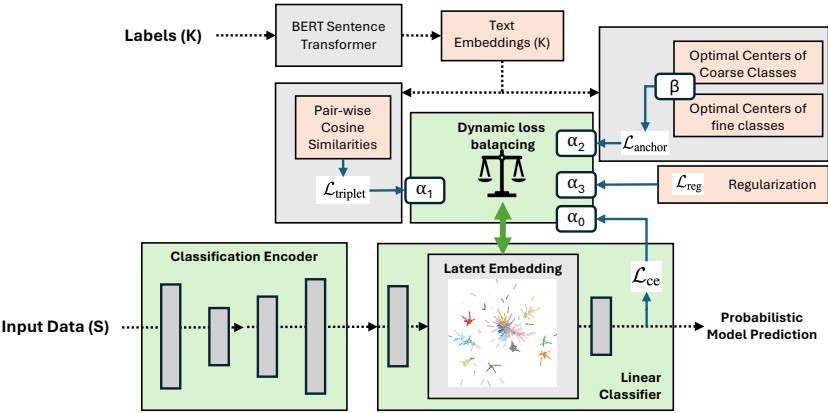

Figure 2: Architecture of LARESA; The latent embedding space is extracted from the model; Latent loss terms (anchor, triplet, regularizer) are combined through a dynamic loss balancing module which learns weights ($\alpha$) to balance the LARESA components with the classic cross-entropy.

## 2 METHOD

Modern classification backbones, while highly effective at distinguishing visual classes, often yield latent spaces that are sparse, fragmented, and lacking semantic structure. Rather than introducing architectural complexity or extensive hyperparameter tuning to mitigate these issues, we propose to enhance representational quality through loss-level supervision, leveraging the already present information in the data, especially in the ground truth labels, addressing the root causes rather than applying post hoc fixes.

The key objective is to orchestrate the latent space by injecting semantic priors directly into the core optimization process, which is commonly based solely on probabilistic cross-entropy. As shown in Figure 2, we interpret pretrained language embeddings as a proxy for external semantic structure, using them to guide visual representations toward more meaningful configurations. By anchoring features to semantic directions and reshaping local neighborhoods in accordance with class-level linguistic similarity, we aim to embed task-relevant information into more meaningful representations.

### 2.1 SEMANTIC SUPERVISION VIA LANGUAGE EMBEDDINGS

Let $x \in \mathbb{R}^{H \times W \times C}$ be an input image and $y \in \{1, \ldots, K\}$ its corresponding class label. A backbone encoder $f_\theta$ maps the input to a latent representation $z = f_\theta(x) \in \mathbb{R}^d$, followed by a linear classifier $\hat{y} = \text{softmax}(Wz)$.

To inject semantic priors into the learning process, we employ the all-MiniLM-L6-v2 (Reimers & Gurevych, 2019b) model from the Sentence-BERT family as our semantic encoder $g_\phi$, offering competitive performance on semantic similarity tasks while maintaining a compact efficient architecture with only 22 million parameters. Unlike CLIP-like methods, **we only activate the SBERT model once for each label** ($K$ times in total) as a preparation step, and is not involved during the training epochs. Given a class name $c_y$, the encoder produces a normalized embedding $e_y = g_\phi(c_y) \in \mathbb{R}^d$ via mean-pooled transformer outputs. These embeddings serve as a fixed semantic scaffold in the latent space, anchoring latent representations to linguistically grounded targets.

### 2.2 ARRANGING LATENT CLUSTERS WITH ABSOLUTE SEMANTIC POSITIONING

We define the cosine anchoring loss $\mathcal{L}_{\text{anchor}}$ as a semantic positioning mechanism that aligns latent representations with language-derived class embeddings at both fine- and coarse-grained levels:

$$\mathcal{L}_{\text{anchor}}(z) = \beta_{\text{fine}} \cdot (1 - \cos(\hat{z}, e_y^{\text{fine}})) + \beta_{\text{coarse}} \cdot (1 - \cos(\hat{z}, e_y^{\text{coarse}})) \tag{1}$$

where $\hat{z} = \frac{z}{\|z\|}$ is the normalized latent feature vector, and $e_y^{\text{fine}}, e_y^{\text{coarse}}$ are Sentence-BERT embeddings for the fine-grained and corresponding coarse label, respectively. The weights $\beta_{\text{fine}}$ and $\beta_{\text{coarse}}$

control the contribution of each semantic level and are derived from a softmax over two learnable parameters, ensuring a convex combination. Alternatively, they can be set as fixed scalars to reflect a predetermined emphasis.

Cosine similarity is chosen as the alignment measure because it captures directional similarity in the embedding space, which is especially suitable for semantic representations derived from language models. Unlike Euclidean distance, cosine similarity disregards magnitude differences and emphasizes semantic orientation. By enforcing latent alignment with both specific and general semantic anchors, this loss encourages latent spaces that respect hierarchical structure. The learnable weighting enables dynamic balancing of semantic abstraction levels throughout training, improving both class separation and generalization without requiring extensive the network architecture modifications.

**LLM-Generated Coarse Labels.**   When coarse label groupings are not available, we can leverage LLMs to automatically infer semantic hierarchies. Given the set of fine-grained class names, we prompt the GPT-4o from OpenAI to produce natural groupings and assign each fine label to a coarse category (OpenAI, 2024). This approach enables semantic hierarchy construction using only the label names, making our method applicable even to datasets without an explicit taxonomy. By integrating both fine and coarse alignment, our method encourages semantically aware latent structure that respects human-level category abstraction, improving generalization and interpretability in downstream tasks. The full list of all dataset label mapping can be found in Appendix C.

## 2.3 MOVING LATENT CLUSTERS WITH RELATIVE SEMANTIC SIMILARITY

To further refine class-wise separation, we apply a semantic triplet loss (Schroff et al., 2015):

$$\mathcal{L}_{\text{triplet}} = \max\left(0, \Delta + d(\hat{z}, e_y) - d(\hat{z}, e_{y^-})\right) \tag{2}$$

where $d(\cdot, \cdot)$ denotes cosine distance, and $y^-$ is a randomly sampled negative label such that $y^- \neq y$. To provide proper efficiency, we decided to randomly select negative samples as opposed to negative hard mining. The margin $\Delta$ controls the minimal separation. We set the margin to $\delta = 0.2$, empirically found to be effective in metric learning settings, which balances informative gradient flow and class separation in the normalized embedding space (Schroff et al., 2015). While $\mathcal{L}_{\text{anchor}}$ serves as a **global positioning constraint** ("where"), the triplet loss acts as a **local repulsion mechanism** ("how far"), pushing clusters away from semantically dissimilar labels and encouraging intra-class compactness.

## 2.4 ADDITIONAL REGULARIZATION

To improve latent space compactness and generalization, we incorporate two standard, class-agnostic geometric constraints, grouped under the joint regularization term $\mathcal{L}_{\text{reg}} = \{\mathcal{L}_{\text{center}}, \mathcal{L}_{\text{var}}\}$, acting as an architecture-independent prior that regularizes feature distributions:

- **Centering loss** $\mathcal{L}_{\text{center}} = \|\mathbb{E}[z]\|^2$, which penalizes latent drift from the origin and encourages symmetric feature distributions (Wen et al., 2016).
- **Variance loss** $\mathcal{L}_{\text{var}} = \sum_i (\sigma_i - 1)^2$, which enforces unit variance across dimensions to stabilize the latent scale (Asperti, 2020).

## 2.5 DYNAMIC LOSS BALANCING

Inspired by related works (Guo et al., 2022; Kendall et al., 2018), we assign a vector of learnable logits $\boldsymbol{\alpha} \in \mathbb{R}^{|\mathcal{L}|}$ to modulate the contribution of each loss term. We consider the loss family

$$\mathcal{L} = \{\mathcal{L}_{\text{ce}}, \mathcal{L}_{\text{anchor}}, \mathcal{L}_{\text{triplet}}, \mathcal{L}_{\text{reg}}\}, \tag{3}$$

corresponding to cross–entropy, cosine anchoring, semantic triplet, and the regularization losses.

**Softmax re-weighting.**   At every training step we map $\boldsymbol{\alpha}$ onto the probability simplex via a softmax,

$$w_i = \frac{\exp(\alpha_i)}{\sum_{j \in \mathcal{L}} \exp(\alpha_j)}, \qquad i \in \mathcal{L}, \tag{4}$$

and form the overall objective as a weighted sum,

$$\mathcal{L}_{\text{total}} = \sum_{i \in \mathcal{L}} w_i \, \mathcal{L}_i. \tag{5}$$

**Optimization benefits.** The softmax guarantees $w_i \geq 0$ and $\sum_i w_i = 1$, preserving loss scale without auxiliary constraints. As $\boldsymbol{\alpha}$ is differentiable, it is updated jointly with the network parameters, enabling the model to adaptively shift its focus among competing objectives as training evolves. This dynamic balancing improves convergence stability and often leads to better generalization in multi-objective learning settings.

**Strategic Visual Interpretation.** We interpret $\mathcal{L}_{\text{anchor}}$ as the *positional guide*, being responsible for projecting features toward their semantic anchors—while $\mathcal{L}_{\text{triplet}}$ modulates the *local geometry* by reinforcing semantic consistency and suppressing category confusion. Their synergy through the weighted sum loss enables both macro-scale semantic grounding and micro-scale refinement in latent space organization.

## 3 Experimental Setup

We conduct a comprehensive evaluation across three model families, namely ResNet50, ResNet101, ResNet152 (He et al., 2016), ConvNeXt-S, ConvNeXt-B, ConvNeXt-L (Liu et al., 2022), and ViT-Small-32, ViT-Base-32, ViT-Large-32 (Dosovitskiy et al., 2021), which are combined with three datasets, CIFAR-100 (Krizhevsky, 2009), CUB-200-2011 (Wah et al., 2011) (abbreviated CUB-200), and TinyImageNet (CS231n) (all initialized with state-of-the-art augmentations (Appendix A)). We setup the ResNet family without weights to train it from scratch, whereas ConvNeXt and ViT models are initialized with ImageNet-1k pretrained weights via `timm` (Wightman, 2019). For each model–dataset pair, we evaluate eight loss configurations involving the combination of the probabilistic cross-entropy, the introduced semantic and distance-based loss functions and the regularization from Appendix 2. Semantic supervision is injected via Sentence-BERT (all-MiniLM-L6-v2) (Reimers & Gurevych, 2019b), mapping the fine- and coarse class names to the embedding space. For CIFAR-100, a set of 20 coarse classes was already available, whereas for Cub-200 and TinyImagenet, we retrieved 28 coarse classes for each through OpenAI GPT4o (OpenAI, 2024) by promoting the LLM to group the list of fine classes. The full mapping of fine to coarse classes can be found in Appendix C. All experiments use a unified PyTorch pipeline with fixed seeds for reproducibility. We applied an early-stopping metric with patience of 100 epochs to prevent overfitting. The code for running and evaluating our experiments is attached in the technical supplements. The full details on our experiment setup are documented in Appendix A.

## 4 Results

To evaluate the impact of semantic alignment and structural regularization, we report results across classification accuracy, latent space compactness, and semantic consistency. Specifically, we analyze four key metrics: top-1 accuracy, silhouette score, nearest-neighbor hit rate (NN-HR), and correlation between latent and semantic distances (Spearman and Pearson). Formal definitions and computation details for all metrics are provided in Appendix B.

### 4.1 Classification Improvement

Table 1 reports top-1 accuracy across a range of ablated loss configurations. Each model–dataset pair is evaluated under combinations of semantic alignment losses and regularization terms, always including cross-entropy as the baseline. Across all architectures and datasets, configurations that incorporate semantic or geometric supervision consistently outperform the cross-entropy baseline, confirming the utility of auxiliary structural guidance in representation learning.

The full LARESA configuration, combining all auxiliary objectives, generally achieves the highest or near-highest accuracy across models and datasets. The same pattern can be found in the complete list of executed models in Appendix D. Performance improvements are consistent, though not strictly monotonic with the addition of more loss terms: in some cases, intermediate configurations

Table 1: Ablation of LARESA for Top-1 accuracy (%) across different loss configurations. The rightmost column shows the improvement of the full LARESA over the CE baseline. Overall, LARESA yields consistent accuracy gains while also reducing training runtime (Runtime statistics in Appendix E). The full set of results across all model families can be found in Appendix D.

| Model/ Dataset | $\mathcal{L}_{ce}$ 
 $\mathcal{L}_{anchor}$ 
 $\mathcal{L}_{triplet}$ 
 $\mathcal{L}_{reg}$ | ✓ 
 – 
 – 
 – | ✓ 
 – 
 – 
 ✓ | ✓ 
 ✓ 
 – 
 – | ✓ 
 ✓ 
 – 
 ✓ | ✓ 
 – 
 ✓ 
 – | ✓ 
 – 
 ✓ 
 ✓ | ✓ 
 ✓ 
 ✓ 
 – | ✓ 
 ✓ 
 ✓ 
 ✓ | Absolute Gain (pp) |
|---|---|---|---|---|---|---|---|---|---|---|
| ResNet50 | CIFAR-100 | 76.55 | 76.38 | 73.18 | 75.49 | 76.88 | 77.25 | 75.72 | **77.49** | +0.9 |
| | CUB-200 | 70.27 | 71.19 | 73.44 | 73.26 | 74.92 | 74.88 | 74.65 | **75.14** | +4.9 |
| | TinyImgNet | 64.55 | 65.70 | 65.55 | 66.05 | 65.45 | **66.20** | 64.65 | 66.15 | +1.6 |
| C.NeXt-B | CIFAR-100 | 72.65 | 74.25 | 73.25 | **74.98** | 73.59 | 73.73 | 74.34 | 74.83 | +2.2 |
| | CUB-200 | 75.94 | 76.38 | 76.05 | 84.04 | 72.04 | 73.25 | **84.24** | 83.33 | +7.4 |
| | TinyImgNet | 65.34 | 65.98 | 66.34 | 66.53 | 66.24 | 66.38 | 66.43 | **66.73** | +1.4 |
| ViT-Large | CIFAR-100 | 63.09 | 64.03 | 64.88 | **65.53** | 64.53 | 64.93 | 64.83 | 65.33 | +2.2 |
| | CUB-200 | 77.14 | 83.03 | 84.03 | 83.83 | 83.73 | 83.23 | 83.33 | **84.13** | +7.0 |
| | TinyImgNet | 62.04 | 64.43 | 64.13 | 64.63 | 64.23 | 64.33 | 64.53 | **64.83** | +2.8 |

slightly outperform the full setup. This suggests that excessive semantic supervision can introduce optimization challenges or lead to over-regularization, particularly in architectures with strong inductive biases. These cases highlight a practical saturation point, where additional structural losses provide diminishing or even adverse returns. Nevertheless, the overall trend is robust—augmenting cross-entropy with semantic and structural losses yields clear benefits, especially in fine-grained or more challenging classification tasks.

Additionally, Table 8 shows that LARESA often reduces the number of epochs needed to reach peak validation accuracy, particularly for ConvNeXt and ViT backbones, while ResNets see more modest gains. This highlights both the efficiency benefits and the architecture-dependent effects of incorporating semantic supervision.

## 4.2 Visual Inspection of the Latent Space

Figure 3 qualitatively illustrates the effect of different loss configurations on the structure of the latent space. We visualize UMAP projections of the ResNet50:CIFAR-100 experiment, colored by the coarse semantic classes. Each subplot corresponds to a distinct loss composition: the vanilla cross-entropy baseline, two partial ablations with either $\mathcal{L}_{anchor}$ or $\mathcal{L}_{triplet}$ combined with regularization $\mathcal{L}_{reg}$, and the full LARESA configuration. The baseline model yields an unstructured embedding with overlapping clusters and poorly separated groups. This reflects its sole focus on discriminative separation without regard for higher-order semantic relationships. Adding anchoring loss induces more compact and outward-separated clusters, suggesting improved alignment with semantic priors. The triplet-based version shows similar gains but with tighter inter-group bindings. The full LARESA configuration displays the most coherent structure: semantically similar groups are well-separated and consistently clustered. For example, categories such as "flowers" and "fruits and vegetables" are not only individually compact but also relatively proximal in latent space, aligning with their underlying linguistic semantics. These observations highlight the role of semantic and structural objectives in shaping meaningful representation geometry. Nevertheless, it is essential to note that dimension reduction through algorithms such as UMAP hinder interpretability since it is a lossy transformation. The remaining experiments' visualizations can be found in Appendix F.

## 4.3 Geometric Compactness and Local Consistency

To evaluate the structural organization of the latent space, we measure two complementary properties: geometric compactness via the Silhouette coefficient (Belyadi & Haghighat, 2021) and local semantic consistency via the nearest-neighbor hit rate (NN-HR), both computed with respect to coarse labels. (Appendix B)

Table 2 shows that the full LARESA configuration substantially improves both metrics across nearly all Model:Dataset pairs. For example, ResNet50:CIFAR-100 increases its silhouette score from

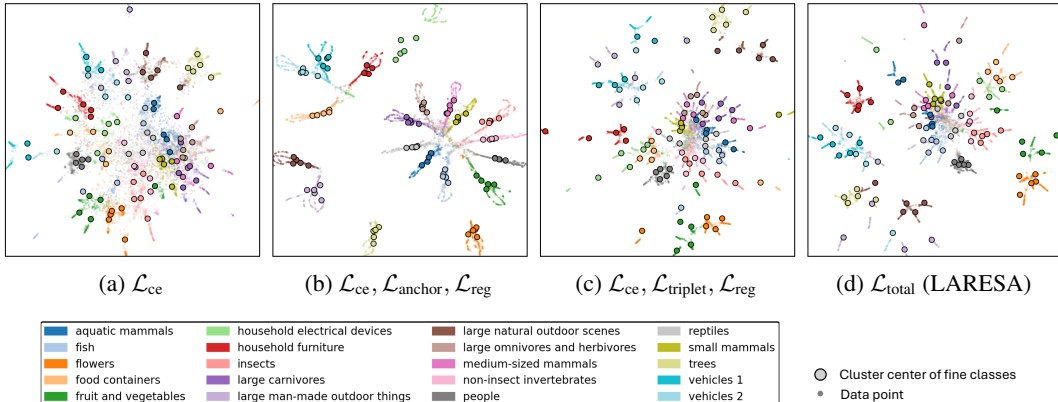

(a) $\mathcal{L}_{ce}$     (b) $\mathcal{L}_{ce}, \mathcal{L}_{anchor}, \mathcal{L}_{reg}$     (c) $\mathcal{L}_{ce}, \mathcal{L}_{triplet}, \mathcal{L}_{reg}$     (d) $\mathcal{L}_{total}$ (LARESA)

Figure 3: Visual comparison of latent space (UMAP) of test set datapoints colorized by coarse labels; plots show the ResNet50:CIFAR-100 experiment; dots with black border constitute fine-class centers; the complete set of visualizations can be found in Appendix F; Note that 2D projections of high dimensional spaces are lossy due to dimension reduction and should be used for qualitative illustration only.

Table 2: Silhouette score (range: –1 to 1) and nearest-neighbor hit rate (NN-HR, %) computed over coarse labels. Comparison of vanilla CE baseline and full LARESA.

| Model | Dataset | Silhouette (CE) | Silhouette (LARESA) | NN-HR (CE) | NN-HR (LARESA) |
|---|---|---|---|---|---|
| ResNet50 | CIFAR-100 | 0.0755 | **0.2476** | 54.00 | **93.00** |
| | CUB-200 | 0.0212 | **0.0704** | 52.00 | **94.50** |
| | TinyImageNet | 0.1023 | **0.2907** | 13.50 | **48.50** |
| ConvNeXt-Base | CIFAR-100 | 0.0875 | **0.4716** | **58.00** | **58.00** |
| | CUB-200 | 0.1426 | **0.4139** | 70.00 | **89.00** |
| | TinyImageNet | 0.0965 | **0.2813** | 16.00 | **47.50** |
| ViT-Large | CIFAR-100 | 0.0256 | **0.1861** | 60.00 | **95.00** |
| | CUB-200 | 0.1836 | **0.2834** | 76.00 | **93.00** |
| | TinyImageNet | 0.0190 | **0.3907** | 12.00 | **43.50** |

0.0755 to 0.2476, and its NN-HR from 54.00% to 93.00%, indicating a tighter and more semantically coherent latent space. Similarly, ViT-Large:TinyImageNet achieves a silhouette score boost from 0.0190 to 0.3907, with NN-HR nearly quadrupling from 12.00% to 43.50%. These improvements reflect that semantic alignment and clustering regularization effectively promote well-separated and internally consistent class clusters. NN-HR particularly emphasizes that latent neighbors increasingly correspond to semantically related classes, which is crucial for generalization under distribution shift or few-shot scenarios.

Even when classification accuracy saturates, the latent geometry continues to benefit from semantic regularization. This is most evident in the CUB-200 results, where ConvNeXt and ViT display large gains in NN-HR despite marginal accuracy differences. This suggests that our method captures structure beyond classification, shaping the embedding space in a way that benefits tasks reliant on meaningful inter-class relationships. These results affirm that LARESA not only enhances class discrimination but also enforces meaningful latent structure, supporting both interpretability and transferability of learned representations.

## 4.4 SEMANTIC CORRELATION OF LATENT SPACE

To assess how well the latent representations preserve semantic relationships, we compute both Spearman rank and Pearson linear correlations between pairwise distances of the models' latent space and the semantic embeddings from the preprocessing through SBERT. This metric reflects the degree to which relational structure among classes is preserved in the learned feature geometry.

Table 3: Spearman and Pearson correlation between semantic and latent distances over coarse labels (range: −1 to 1). Comparison of vanilla CE baseline and full LARESA.

| Model | Dataset | Spearman (CE) | Spearman (LARESA) | Pearson (CE) | Pearson (LARESA) |
|-------|---------|---------------|-------------------|--------------|------------------|
| ResNet50 | CIFAR-100 | 0.3174 | **0.5872** | 0.3995 | **0.5916** |
| | CUB-200 | 0.0463 | **0.2977** | 0.1238 | **0.4106** |
| | TinyImageNet | -0.0065 | **0.0053** | 0.0002 | **0.0073** |
| ConvNeXt-Base | CIFAR-100 | 0.3025 | **0.7718** | 0.3916 | **0.8340** |
| | CUB-200 | 0.0214 | **0.5854** | 0.1244 | **0.6952** |
| | TinyImageNet | 0.0256 | **0.6240** | 0.0326 | **0.6455** |
| ViT-Large | CIFAR-100 | 0.2919 | **0.5870** | 0.3961 | **0.6318** |
| | CUB-200 | 0.0750 | **0.5414** | 0.1830 | **0.6586** |
| | TinyImageNet | 0.0284 | **0.7826** | 0.0349 | **0.8164** |

Table 3 reports the correlation values computed over coarse class centroids, comparing the cross-entropy baseline with the full LARESA configuration. Across all datasets and architectures, both Spearman and Pearson correlations consistently improve with LARESA, indicating stronger alignment between the learned visual representations and the semantic class structure. This is particularly evident for ViT-Large:TinyImageNet, reflecting a shift from almost no semantic ordering to highly aligned relational structure in the latent space. These results confirm that LARESA effectively integrates linguistic priors into the geometric arrangement of class representations by utilizing the anchor loss as a positional guidance and the triplet loss as a force to achieve the semantic alignment. Beyond the previously presented performance gain, LARESA promotes latent interpretability and coherence, enabling models to generalize across semantically related classes.

### 4.5 AUTONOMOUS PRIORITY ORGANIZATION

Figure 4 visualizes the evolution of $\alpha$ weights over training epochs for ViT-Large:CIFAR-100. The remaining plots for all other model–dataset combinations are included in Appendix G. We observed for most runs the alpha evolution follow this case: the weights of cross-Entropy $\mathcal{L}_{ce}$ and variance start to decrease from the beginning, followed by two peaks of anchoring loss and triplet loss, with the anchoring loss first and triplet later. Then the weight of centering loss gradually increases close to 1, and that of every other loss degrades close to zero.

Recalling Figure 1, we could explain this from the information perspective: $\mathcal{L}_{ce}$ holds the discrete class index distribution information. When the training dataset probability metrics reach high levels, $\mathcal{L}_{ce}$ can be considered fully ex-

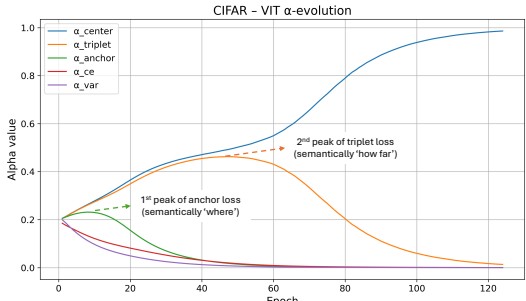

Figure 4: Evolution of loss term weights $\alpha_i$ for ViT-Large:CIFAR-100; each curve reflects the learned importance of a loss component; the complete list set of loss weight can be found in Appendices G and H.

ploited and further optimization will likely lead to overfitting. In the next phase $\mathcal{L}_{anchor}$ and $\mathcal{L}_{triplet}$ are more prioritized for semantic alignment, which further improves the generalization from training data to unseen data. $\mathcal{L}_{anchor}$ first puts the clusters at absolute positions ("where"), then $\mathcal{L}_{triplet}$ expels the clusters away from each other according to semantic dissimilarity ("how far"). When the clusters are well semantically aligned, $\mathcal{L}_{anchor}$ and $\mathcal{L}_{triplet}$ would have also been fully exploited. $\mathcal{L}_{center}$ on the other hand is only a geometric structuring regularizer and does not contain any specific information (e.g. probability distribution or semantic partition), and can be indefinitely optimized to infinity. The coefficients balancing fine and coarse anchoring losses (Appendix H) also consistently show a self-organizing semantic **hierarchy focus** shift: the coarse classes are prioritized at first, often with a peak, then the priority of fine classes takes over. Notably these behaviors are consistent in most Model:Dataset combinations, entirely emerging from the automatic training optimization process.

## 5 RELATED WORK

**Semantic Alignment.** A long line of work has explored bridging visual and semantic spaces, from zero-shot and few-shot recognition (Rahman et al., 2020; Hao et al., 2021) to large vision–language models (Radford et al., 2021; Singh et al., 2022; Chen et al., 2024b; Wang et al., 2024; Chen et al., 2024a). CLIP (Radford et al., 2021), for example, establishes alignment through large-scale contrastive pretraining, while follow-ups such as DRESS (Chen et al., 2024b) and SERL (Huang et al., 2025) refine this alignment through natural-language feedback or explicit semantic hierarchies. Other approaches integrate prompts (Sun et al., 2024), causal associations (Meng et al., 2025), or BERT embeddings (Yan et al., 2021; 2022; Li et al., 2022) to inject semantic priors. Unlike these works, which often depend on architectural changes, external networks, or massive pretraining, LARESA contributes a lightweight, loss-based mechanism that applies directly to standard classifiers. Rather than fusing modalities at the input level, we impose semantic structure directly in the latent space via anchor and triplet objectives derived from label embeddings.

**Metric Learning for Latent Optimization.** Metric learning methods shape latent distances to align with downstream objectives, e.g., CoBO and TaskMet (Lee et al., 2023; Bansal et al., 2023) for optimization tasks, sparse Bayesian models for generalization (Zabihzadeh et al., 2019), or magnitude-based diversity metrics to stabilize generative modeling (Limbeck et al., 2024; Samuel et al., 2023; Vepa et al., 2024). These works highlight the utility of explicit distance structuring, but are typically domain-specific or tuned to specialized optimization problems. LARESA differs by embedding semantic relations derived from language models into the latent metric itself, thereby generalizing across domains and architectures without retraining regimes tailored to each task.

**Latent Space Structuring.** Representation learning has long investigated structured latent spaces for disentanglement and controllability (Trunz et al., 2022; Marsot et al., 2022). Generative approaches (VAEs, GANs) deliberately enforce structure (Hu et al., 2024; Xu et al., 2024), while others map latent axes to interpretable semantic factors (Liu et al., 2019). Hierarchical correlation measures such as CPCC (Zeng et al., 2022; Sinha et al., 2024) embed taxonomy-like structure. LARESA extends this direction by coupling semantic embeddings with geometric regularizers: anchors align latent centroids with SBERT directions, triplets enforce semantic distances, and variance regularization ensures compactness, yielding structured yet task-specific representations without modifying backbone architectures.

**Loss Design for Interpretability.** Ante-hoc interpretability strategies design loss functions that enforce human-understandable constraints, as opposed to post-hoc explanations. Examples include harmonic losses (Baek et al., 2025), feature-preserving penalties in medical imaging (Dong & Basu, 2024), or concept distillation losses (Garouani et al., 2024). Even interactive approaches like HILL (Geissler et al., 2025) inject human control directly into latent optimization. LARESA shares this spirit of embedding interpretability into training, but focuses on scalable, automated alignment through language priors rather than explicit human labeling or constraints. This positions our work as a general-purpose bridge between interpretability losses and semantic alignment in supervised classification.

## 6 CONCLUSION

We investigated the role of semantic structure in improving generalization for high-class image classification. LARESA injects existing or even LLM-derived semantic priors into untouched models via auxiliary losses, avoiding expensive pretraining or architectural changes. Unlike prior sample-wise contrastive methods, our framework enforces cluster-level coherence through adaptive weighting of geometric objectives alongside cross-entropy. Experiments across our set of models and datasets show consistent accuracy gains, more coherent latent clusters, and improved metrics such as silhouette score and hierarchy alignment. Further implications, limitations, and extensions beyond classification are discussed in Appendix I. To sum up, LARESA offers a lightweight way to retrofit existing classifiers with semantic priors, bridging the gap between human semantics and machine representations.

## REPRODUCIBILITY STATEMENT

We ensure reproducibility by providing full implementation details, including preprocessing steps, hyperparameters, model architectures, and training schedules in the Supplementary Material and Appendix. All datasets used (CIFAR-100, CUB-200, TinyImageNet) are publicly available, and we specify our validation/test splits to enable direct comparison. Extended results across architectures, runtime statistics, and fine-to-coarse label mappings are included in the appendix as well.

## ETHICS STATEMENT

This work uses only publicly available benchmark datasets without personal or sensitive information. The semantic embeddings employed (e.g., SBERT and GPT-4o-based mappings) may reflect biases inherent to large language models and text corpora. To mitigate this, we manually verified label mappings and highlight the importance of using domain-specific embeddings in sensitive contexts (e.g., medical or legal applications). We believe the benefits of improved training efficiency and generalization outweigh the limited ethical risks, but encourage practitioners to carefully consider potential downstream bias propagation.

## LARGE LANGUAGE MODEL (LLM) USAGE

In line with ICLR 2026 requirements, we disclose all uses of large language models (LLMs) during the preparation of this work. LLMs were used in three ways:

(1) Writing and polishing. LLMs assisted in refining the clarity, readability, and grammar of the manuscript. They did not generate new ideas, methods, or results, but helped ensure that the presentation was precise and accessible. The scientific content, argumentation, and contributions remain entirely authored by us.

(2) Retrieval and discovery. LLMs were used as a supporting tool to accelerate literature search and organization of related work, particularly to surface recent references and structure them into coherent categories. All cited works were independently verified and integrated by the authors, ensuring accuracy and completeness of the survey.

(3) Fine-to-coarse label mappings. For CUB-200 and TinyImageNet, we employed GPT-4o to propose semantically coherent groupings of fine-grained labels into higher-level categories. These suggestions served as a starting point; we manually reviewed and validated all mappings to guarantee interpretability and uniqueness. This use of LLMs accelerated a tedious clustering task but did not replace human judgment.

Overall, LLMs were used only as auxiliary tools to improve presentation, efficiency, and data preparation. They did not influence the design of our method, the implementation of experiments, or the interpretation of results, and therefore do not compromise the originality, validity, or reproducibility of our research.

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

# A EXPERIMENTAL SETUP

We evaluate LARESA across 29 experimental configurations in total, combining three backbone families, ResNet50, ResNet101, ResNet152 (He et al., 2016), ConvNeXt-S, ConvNeXt-B, ConvNeXt-L (Liu et al., 2022), and ViT-Small-32, ViT-Base-32, ViT-Large-32 (Dosovitskiy et al., 2021), with three visual classification datasets of increasing difficulty: CIFAR-100 (Krizhevsky, 2009), CUB-200-2011 (Wah et al., 2011), and TinyImageNet (CS231n).

**Model Initialization.** We train each ResNet model from scratch without pretrained weights to assess generalization from a cold start. ConvNeXt and ViT models are initialized with ImageNet-1k pretrained weights via the `timm` library (Wightman, 2019).

**Semantic Embedding Preprocessing.** We incorporate semantic supervision by embedding the fine-grained and coarse class labels using Sentence-BERT (all-MiniLM-L6-v2) (Reimers & Gurevych, 2019b). This model generates fixed-size 384-dimensional sentence embeddings for each class name or description. To enable alignment between image features and language embeddings, we adapt the model's classification head so that the penultimate latent feature vector also resides in $\mathbb{R}^{384}$. This allows auxiliary semantic losses (e.g., cosine alignment, triplet margin) to be computed directly between the model's latent representation and the corresponding semantic embedding. Notably, this adjustment does not affect the architecture of the backbone network and preserves compatibility with pretrained weights.

**Fine-to-Coarse Mapping.** CIFAR-100 includes 20 coarse labels natively. For CUB-200 and TinyImageNet, which lack hierarchical groupings, we derive 28 coarse categories via OpenAI GPT-4o (OpenAI, 2024) by prompting the model to cluster the fine classes into semantically coherent groups. These coarse mappings are used in evaluation and loss supervision and are provided in Appendix C.

**Loss Configuration.** Each model–dataset pair is trained under eight ablation configurations combining the standard cross-entropy loss with auxiliary semantic and structural objectives: cosine anchoring ($\mathcal{L}_{\text{anchor}}$), triplet margin ($\mathcal{L}_{\text{triplet}}$), center loss ($\mathcal{L}_{\text{center}}$), and variance regularization ($\mathcal{L}_{\text{variance}}$). Loss weights are dynamically modulated via learnable coefficients $\alpha$ (per-loss scaling) and $\beta$ (fine/coarse balancing), initialized to 1 and optimized jointly with model parameters.

**Optimization Details.** All models are optimized using Adam (no weight decay) with learning rates of $1 \times 10^{-3}$ for ResNet and $3 \times 10^{-4}$ for ConvNeXt and ViT. Models are trained for up to 1000 epochs with early stopping based on validation accuracy (patience = 100 epochs). No learning rate schedule is applied.

**Data Preprocessing and Augmentation.** All datasets are preprocessed with autoAugment (Cubuk et al., 2019) and the following dataset-specific transforms:

- **CIFAR-100:** `RandomCrop(32, padding=4)`, `RandomHorizontalFlip()`, `ToTensor()`.
- **CUB-200:** `CenterCrop(224)`, `RandomHorizontalFlip()`, `ImageNet normalization`.
- **TinyImageNet:** Training uses `RandomResizedCrop(64)` and `RandomHorizontalFlip`; evaluation uses `Resize(64)` and `CenterCrop(64)`. Both use ImageNet-style normalization.

For all datasets, we split the provided test sets into 50/50 validation/test partitions using a fixed seed.

**Random Seed and Reproducibility.** To ensure strict reproducibility of all experiments, we fix the global random seed to `SEED=42` across all components of the pipeline. This includes NumPy, Python's `random`, and PyTorch's CPU and CUDA random number generators. We enforce deterministic behavior by setting `torch.backends.cudnn.deterministic=True`, disabling CUDNN benchmarking, and explicitly enabling deterministic algorithms with

`torch.use_deterministic_algorithms(True, warn_only=True)`. For dataloading, we pass a seeded `torch.Generator` to each `DataLoader` instance and use a custom `seed_worker` function to propagate consistent seeds to each worker process.

## A.1 Hardware and Runtime

All models are trained on a shared SLURM-job compute cluster equipped with NVIDIA A100 and A6000 GPUs. Each experiment runs for approximately 2–8 hours depending on dataset size, model complexity, and loss configuration. Each experiment required approximately 8–14 GB of GPU memory.

**Logging and Evaluation.** We log all training metrics—including loss values, dynamic $(\alpha/\beta)$ weights, and classification performance—to both CSV and Weights & Biases. Latent features are projected and saved at regular intervals for UMAP-based visualization. Evaluation includes top-1 accuracy, silhouette score, nearest-neighbor hit rate (NN-HR), and Spearman/Pearson correlations between semantic and latent distances.

**Code Availability.** The complete codebase, including training scripts, evaluation utilities, data loaders, loss implementations, and configuration files, is provided in the technical supplements to facilitate full reproducibility.

## B Latent Space Evaluation Metrics

To evaluate the quality and structure of the learned latent representations, we employ five complementary metrics: top-1 classification accuracy, nearest-neighbor hit rate, silhouette score, and the correlation between semantic and latent class centers via Spearman and Pearson coefficients. Together, these metrics provide a holistic view of both discriminative performance and latent geometry.

**Top-1 Accuracy.** This standard metric measures the fraction of samples for which the predicted class corresponds to the ground-truth label:

$$\text{Top-1 Accuracy} = \frac{1}{N} \sum_{i=1}^{N} \mathbf{1} \left[ \hat{y}_i = y_i \right] \tag{6}$$

**Nearest-Neighbor Hit Rate (NN-Hit).** We define a coarse-level hit rate to assess whether the nearest latent class centroid corresponds to a semantically related (coarse) class. Let $\mu_i$ denote the centroid of class $i$ in latent space, and $c(i)$ its associated coarse label. The hit rate is computed as:

$$\text{NN-Hit} = \frac{1}{K} \sum_{i=1}^{K} \mathbf{1} \left[ c(i) = c(j^*) \right], \quad j^* = \arg\min_{j \neq i} \| \mu_i - \mu_j \|_2 \tag{7}$$

where $K$ is the number of predicted classes and $j^*$ is the nearest neighbor of $i$ (excluding $i$ itself). This metric measures coarse-grained alignment between predicted class clusters.

**Silhouette Score.** The silhouette score quantifies cluster compactness and separation:

$$s(i) = \frac{b(i) - a(i)}{\max\{a(i), b(i)\}}, \quad \text{Silhouette} = \frac{1}{N} \sum_{i=1}^{N} s(i) \tag{8}$$

where $a(i)$ is the mean intra-class distance for sample $i$, and $b(i)$ is the mean distance to the nearest different class. Values range from $-1$ (poor separation) to $+1$ (well-separated clusters).

**Spearman Rank Correlation.** To measure global alignment between the latent and semantic structures, we compute pairwise distances between class centroids in latent space ($D_{\text{lat}}$) and corresponding language embeddings ($D_{\text{sem}}$), and evaluate their monotonic correlation:

$$\rho = \text{Spearman}(D_{\text{lat}}, D_{\text{sem}}) = \text{corr}(\text{rank}(D_{\text{lat}}), \text{rank}(D_{\text{sem}})) \tag{9}$$

This captures whether the ordering of semantic distances is preserved in the learned space.

**Pearson Correlation.** Complementarily, we report the Pearson correlation between the same distance matrices:

$$r = \text{Pearson}(D_{\text{lat}}, D_{\text{sem}}) = \frac{\text{cov}(D_{\text{lat}}, D_{\text{sem}})}{\sigma_{D_{\text{lat}}} \sigma_{D_{\text{sem}}}} \tag{10}$$

Unlike Spearman, Pearson is sensitive to absolute scaling and measures linear alignment between latent and semantic distances.

Together, these metrics offer a quantitative view into the structural and semantic quality of the latent space beyond classification accuracy.

## C  FINE-TO-COARSE LABEL MAPPINGS

We construct a mapping function

$$\texttt{fine2coarse} : \mathcal{Y}_{\text{fine}} \to \mathcal{Y}_{\text{coarse}},$$

where each fine label $y \in \mathcal{Y}_{\text{fine}}$ is assigned to a coarse label in $\mathcal{Y}_{\text{coarse}}$. These mappings are used to obtain Sentence-BERT embeddings at two hierarchy levels of semantic abstraction (Appendix 2), and to compute coarse-grained metrics such as the nearest-neighbor hit rate.

**Datasets.** For **CIFAR-100**, we directly adopt the official taxonomy, where 100 fine-grained categories are grouped into 20 superclasses. This ensures comparability with prior work that leverages hierarchical information.

For **CUB-200-2011** and **TinyImageNet**, no official coarse grouping exists. To establish a semantically meaningful structure, we generated mappings using a large language model (OpenAI GPT-4o) (OpenAI, 2024). We prompted the model with the full list of fine-grained labels and asked it to cluster them into a manageable set of superclasses based on shared semantic or functional attributes (e.g., grouping bird species by family or habitat, or grouping ImageNet categories into "vehicles," "instruments," or "animals"). The output was post-processed to ensure that: (i) every fine label was uniquely assigned, (ii) groups were balanced in size, and (iii) the assigned superclass names were semantically interpretable and aligned with the label set. When necessary, we merged overly specific groups or corrected inconsistent mappings manually.

**Rationale.** The motivation for using LLM-based mappings is twofold: (i) it enables the construction of hierarchical semantic anchors even for datasets without predefined taxonomies, and (ii) it provides a scalable and reproducible procedure for deriving coarse-level abstractions across domains. Importantly, because LARESA operates on continuous SBERT embeddings, imperfect or noisy groupings degrade gracefully, whereas the model learns to down-weight misaligned anchors via dynamic loss balancing (see Section 2).

Table 4: Mapping of CIFAR-100

| Coarse Label | Fine Labels |
|---|---|
| aquatic mammals | beaver, dolphin, otter, seal, whale |
| fish | aquarium_fish, flatfish, ray, shark, trout |
| flowers | orchid, poppy, rose, sunflower, tulip |
| food containers | bottle, bowl, can, cup, plate |
| fruit and vegetables | apple, mushroom, orange, pear, sweet_pepper |
| household electrical devices | clock, keyboard, lamp, telephone, television |
| household furniture | bed, chair, couch, table, wardrobe |
| insects | bee, beetle, butterfly, caterpillar, cockroach |
| large carnivores | bear, leopard, lion, tiger, wolf |
| large man-made outdoor things | bridge, castle, house, road, skyscraper |
| large natural outdoor scenes | cloud, forest, mountain, plain, sea |
| large omnivores and herbivores | camel, cattle, chimpanzee, elephant, kangaroo |
| medium-sized mammals | fox, porcupine, possum, raccoon, skunk |
| non-insect invertebrates | crab, lobster, snail, spider, worm |
| people | baby, boy, girl, man, woman |
| reptiles | crocodile, dinosaur, lizard, snake, turtle |
| small mammals | hamster, mouse, rabbit, shrew, squirrel |
| trees | maple_tree, oak_tree, palm_tree, pine_tree, willow_tree |
| vehicles 1 | bicycle, bus, motorcycle, pickup_truck, train |
| vehicles 2 | lawn_mower, rocket, streetcar, tank, tractor |

Table 5: Mapping of CUB-200-2011

| Coarse Label | Fine Labels |
|---|---|
| Blackbirds-Grackles | Baltimore-Oriole, Boat-tailed-Grackle, Bobolink, Brewer-Blackbird, Bronzed-Cowbird, Hooded-Oriole, Orchard-Oriole, Red-winged-Blackbird, Rusty-Blackbird, Scott-Oriole, Shiny-Cowbird, Western-Meadowlark, Yellow-headed-Blackbird |
| Crows-Jays | American-Crow, Blue-Jay, Clark-Nutcracker, Common-Raven, Fish-Crow, Florida-Jay, Green-Jay, White-necked-Raven |
| Cuckoos-Nightjars | Black-billed-Cuckoo, Chuck-will-Widow, Geococcyx, Groove-billed-Ani, Mangrove-Cuckoo, Nighthawk, Whip-poor-Will, Yellow-billed-Cuckoo |
| Diving-Seabirds | Brandt-Cormorant, Crested-Auklet, Horned-Puffin, Least-Auklet, Pacific-Loon, Parakeet-Auklet, Pelagic-Cormorant, Pigeon-Guillemot, Red-faced-Cormorant, Rhinoceros-Auklet |
| Ducks-Geese | Gadwall, Hooded-Merganser, Mallard, Red-breasted-Merganser |
| Finches-Buntings | American-Goldfinch, Blue-Grosbeak, Cardinal, European-Goldfinch, Evening-Grosbeak, Gray-crowned-Rosy-Finch, Indigo-Bunting, Lazuli-Bunting, Painted-Bunting, Pine-Grosbeak, Purple-Finch, Rose-breasted-Grosbeak |
| Flycatchers | Acadian-Flycatcher, Gray-Kingbird, Great-Crested-Flycatcher, Least-Flycatcher, Olive-sided-Flycatcher, Sayornis, Scissor-tailed-Flycatcher, Tropical-Kingbird, Vermilion-Flycatcher, Western-Wood-Pewee, Yellow-bellied-Flycatcher |
| Grebes | Eared-Grebe, Horned-Grebe, Pied-billed-Grebe, Western-Grebe |
| Gulls-Terns | Artic-Tern, Black-Tern, California-Gull, Caspian-Tern, Common-Tern, Elegant-Tern, Forsters-Tern, Glaucous-winged-Gull, Heermann-Gull, Herring-Gull, Ivory-Gull, Least-Tern, Long-tailed-Jaeger, Pomarine-Jaeger, Red-legged-Kittiwake, Ring-billed-Gull, Slaty-backed-Gull, Western-Gull |
| Hummingbirds | Anna-Hummingbird, Green-Violetear, Ruby-throated-Hummingbird, Rufous-Hummingbird |
| Kingfishers | Belted-Kingfisher, Green-Kingfisher, Pied-Kingfisher, Ringed-Kingfisher, White-breasted-Kingfisher |
| Large-Seabirds | Black-footed-Albatross, Brown-Pelican, Frigatebird, Laysan-Albatross, Northern-Fulmar, Sooty-Albatross, White-Pelican |
| Larks | Horned-Lark |
| Mockingbirds-Thrashers | Brown-Thrasher, Mockingbird, Sage-Thrasher |
| Nuthatches | White-breasted-Nuthatch |
| Pipits-Wagtails | American-Pipit |
| Shrikes | Great-Grey-Shrike, Loggerhead-Shrike |
| Songbirds | Gray-Catbird, Spotted-Catbird, Yellow-breasted-Chat |
| Sparrows-Allies | Baird-Sparrow, Black-throated-Sparrow, Brewer-Sparrow, Chipping-Sparrow, Clay-colored-Sparrow, Dark-eyed-Junco, Eastern-Towhee, Field-Sparrow, Fox-Sparrow, Grasshopper-Sparrow, Green-tailed-Towhee, Harris-Sparrow, Henslow-Sparrow, House-Sparrow, Le-Conte-Sparrow, Lincoln-Sparrow, Nelson-Sharp-tailed-Sparrow, Savannah-Sparrow, Seaside-Sparrow, Song-Sparrow, Swamp-Sparrow, Tree-Sparrow, Vesper-Sparrow, White-crowned-Sparrow, White-throated-Sparrow |
| Starlings | Cape-Glossy-Starling |
| Swallows | Bank-Swallow, Barn-Swallow, Cliff-Swallow, Tree-Swallow |
| Tanagers | Scarlet-Tanager, Summer-Tanager |
| Treecreepers | Brown-Creeper |
| Vireos | Black-capped-Vireo, Blue-headed-Vireo, Philadelphia-Vireo, Red-eyed-Vireo, Warbling-Vireo, White-eyed-Vireo, Yellow-throated-Vireo |
| Warblers | American-Redstart, Bay-breasted-Warbler, Black-and-white-Warbler, Black-throated-Blue-Warbler, Blue-winged-Warbler, Canada-Warbler, Cape-May-Warbler, Cerulean-Warbler, Chestnut-sided-Warbler, Common-Yellowthroat, Golden-winged-Warbler, Hooded-Warbler, Kentucky-Warbler, Louisiana-Waterthrush, Magnolia-Warbler, Mourning-Warbler, Myrtle-Warbler, Nashville-Warbler, Northern-Waterthrush, Orange-crowned-Warbler, Ovenbird, Palm-Warbler, Pine-Warbler, Prairie-Warbler, Prothonotary-Warbler, Swainson-Warbler, Tennessee-Warbler, Wilson-Warbler, Worm-eating-Warbler, Yellow-Warbler |
| Waxwings | Bohemian-Waxwing, Cedar-Waxwing |
| Woodpeckers | American-Three-toed-Woodpecker, Downy-Woodpecker, Northern-Flicker, Pileated-Woodpecker, Red-bellied-Woodpecker, Red-cockaded-Woodpecker, Red-headed-Woodpecker |
| Wrens | Bewick-Wren, Cactus-Wren, Carolina-Wren, House-Wren, Marsh-Wren, Rock-Wren, Winter-Wren |

Table 6: Mapping of TinyImageNet

| Coarse Label | Fine Labels |
|---|---|
| amphibians | European-fire-salamander, bullfrog, tailed-frog |
| aquatic-animals | barracouta, brain-coral, chambered-nautilus, chiton, coho-salmon, goldfish, jellyfish, pufferfish, rock-beauty, sea-cucumber, sea-slug, snail |
| arthropods-other | black-and-gold-garden-spider, centipede, scorpion, tarantula, trilobite |
| bears-pandas | American-black-bear, bear, brown-bear, giant-panda |
| birds | albatross, bustard, dowitcher, goose, king-penguin, oystercatcher, pelican, red-backed-sandpiper, redshank, white-stork |
| canids | Chihuahua, English-foxhound, German-shepherd, Labrador-retriever, Walker-hound, golden-retriever, redbone |
| clothing | apron, bib, bikini, bonnet, bow-tie, brassiere, breastplate, cardigan, fur-coat, jean, jersey, kimono, knee-pad, lab-coat, miniskirt, mitten, neck-brace |
| containers | barrel, bottlecap, bucket |
| electronic-devices | CD-player, cellular-phone, computer-keyboard, home-theatre, iPod, joystick, laptop, loudspeaker |
| felids | Egyptian-cat, Persian-cat, cougar, lion, tabby-cat, tiger |
| furniture-fixtures | bookcase, chest, desk, dining-table, lampshade, pedestal |
| hoofed-mammals | Arabian-camel, bison, gazelle, zebra |
| household-items | ashcan, backpack, bathtub, binder, broom, candle, iron, jack-o-lantern |
| insects | bee, cockroach, dragonfly, grasshopper, ladybug, mantis, monarch-butterfly, sulphur-butterfly, walking-stick |
| kitchenware-tableware | beaker, beer-bottle, beer-glass, caldron, espresso-maker, frying-pan, goblet, ladle, mixing-bowl |
| large-herbivores | African-elephant, Indian-elephant, hippopotamus |
| marine-mammals | dugong, grey-whale, killer-whale, sea-lion |
| musical-instruments | accordion, brass, cello, chime, electric-guitar, guitar, oboe, organ |
| personal-items | face-powder, jigsaw-puzzle, lipstick, oxygen-mask, pencil-box, perfume |
| primates | baboon, chimpanzee |
| reptiles | American-alligator, boa-constrictor |
| small-mammals | guinea-pig, koala, wood-rabbit |
| sports-equipment | baseball, basketball, drumstick, dumbbell, golf-ball, horizontal-bar |
| structures-places | apiary, bannister, barbershop, barn, beam, bell-cote, birdhouse, boathouse, bookshop, breakwater, butcher-shop, cliff-dwelling, confectionery, dam, flagpole, fountain, lumbermill, maypole, maze, obelisk |
| tools-devices | barometer, binoculars, cash-machine, chain, crane, hourglass, knot, loupe, magnetic-compass, nail, pencil-sharpener |
| vehicles-land | barrow, beach-wagon, bullet-train, cab, car-mirror, convertible, electric-locomotive, freight-car, go-kart, golfcart, horse-cart, jeep, minivan, moving-van, oxcart, tandem-bicycle |
| vehicles-water | canoe, container-ship, gondola |
| weapons | assault-rifle, bow, cannon, gasmask, missile |

# D FULL ABLATION TOP-1 ACCURACY

Table 7: Ablation of LARESA for Top-1 accuracy (%) across different loss configurations. The rightmost column shows the absolute improvement (percentage points) of the full LARESA over the CE baseline.

| Model/ Dataset | $\mathcal{L}_{ce}$ ✓ $\mathcal{L}_{anchor}$ − $\mathcal{L}_{triplet}$ − $\mathcal{L}_{reg}$ − | ✓ − − ✓ | ✓ ✓ − − | ✓ ✓ − ✓ | ✓ − ✓ − | ✓ − ✓ ✓ | ✓ ✓ ✓ − | ✓ ✓ ✓ ✓ | Absolute Gain (pp) |
|---|---|---|---|---|---|---|---|---|---|
| ResNet50 | CIFAR-100 | 76.55 | 76.38 | 73.18 | 75.49 | 76.88 | 77.25 | 75.72 | **77.49** | +0.9 |
| | CUB-200 | 70.27 | 71.19 | 73.44 | 73.26 | 74.92 | 74.88 | 74.65 | **75.14** | +4.9 |
| | TinyImgNet | 64.55 | 65.70 | 65.55 | 66.05 | 65.45 | **66.20** | 64.65 | 66.15 | +1.6 |
| ResNet101 | CIFAR-100 | 77.15 | 78.02 | 74.33 | 76.93 | 77.48 | 77.83 | 76.15 | **78.08** | +0.9 |
| | CUB-200 | 72.94 | 73.89 | 75.63 | 75.77 | 76.01 | 76.28 | 75.72 | **76.57** | +3.6 |
| | TinyImgNet | 65.94 | 66.83 | 66.68 | 67.02 | 66.38 | 66.63 | 66.24 | 66.98 | +1.0 |
| ResNet152 | CIFAR-100 | 77.44 | 78.15 | 74.94 | 77.05 | 77.64 | 78.08 | 77.24 | **78.33** | +0.9 |
| | CUB-200 | 73.69 | 74.54 | 75.64 | 76.09 | 76.26 | 76.34 | 76.19 | **76.75** | +3.1 |
| | TinyImgNet | 66.44 | 67.18 | 66.88 | 67.28 | 66.83 | 66.98 | 66.88 | **67.43** | +1.0 |
| ConvNeXt-S | CIFAR-100 | 72.31 | **74.83** | 73.45 | 74.63 | 73.13 | 73.29 | 72.55 | 73.89 | +1.6 |
| | CUB-200 | 74.82 | 75.96 | 70.33 | **83.14** | 71.89 | 72.62 | 82.81 | 82.23 | +7.4 |
| | TinyImgNet | 65.25 | 65.43 | 65.88 | 66.02 | 65.79 | 65.83 | 66.14 | 66.33 | +1.1 |
| ConvNeXt-B | CIFAR-100 | 72.65 | 74.25 | 73.25 | **74.98** | 73.59 | 73.73 | 74.34 | 74.83 | +2.2 |
| | CUB-200 | 75.94 | 76.38 | 76.05 | 84.04 | 72.04 | 73.25 | **84.24** | 83.33 | +7.4 |
| | TinyImgNet | 65.34 | 65.98 | 66.34 | 66.53 | 66.24 | 66.38 | 66.43 | **66.73** | +1.4 |
| ConvNeXt-L | CIFAR-100 | 72.84 | 74.44 | 73.45 | **75.18** | 73.74 | 73.93 | 74.49 | 74.88 | +2.0 |
| | CUB-200 | 76.24 | 77.14 | 77.84 | 84.24 | 74.04 | 75.04 | **84.34** | 83.94 | +7.7 |
| | TinyImgNet | 65.49 | 66.13 | 66.24 | 66.43 | 66.34 | 66.38 | 66.54 | **66.83** | +1.3 |
| ViT-Small-32 | CIFAR-100 | 61.14 | 62.09 | **63.48** | 62.83 | 62.04 | 62.54 | 62.14 | 62.78 | +1.6 |
| | CUB-200 | 74.81 | 83.43 | 77.85 | 82.81 | 83.69 | 83.06 | 64.89 | **83.91** | +9.1 |
| | TinyImgNet | 60.94 | **64.03** | 62.28 | 63.23 | 61.83 | 62.53 | 63.04 | 63.83 | +2.9 |
| ViT-Base-32 | CIFAR-100 | 62.28 | 63.23 | 64.18 | **64.83** | 63.73 | 64.13 | 64.03 | 64.63 | +2.3 |
| | CUB-200 | 76.04 | 84.13 | 78.07 | 84.63 | 84.53 | 84.03 | 84.23 | **85.03** | +9.0 |
| | TinyImgNet | 61.54 | 63.83 | 63.13 | 63.88 | 63.28 | 63.73 | 63.63 | **64.03** | +2.5 |
| ViT-Large-32 | CIFAR-100 | 63.09 | 64.03 | 64.88 | **65.53** | 64.53 | 64.93 | 64.83 | 65.33 | +2.2 |
| | CUB-200 | 77.14 | 83.03 | 84.03 | 83.83 | 83.73 | 83.23 | 83.33 | **84.13** | +7.0 |
| | TinyImgNet | 62.04 | 64.43 | 64.13 | 64.63 | 64.23 | 64.33 | 64.53 | **64.83** | +2.8 |

# E  RUNTIME

Table 8: Convergence runtime measured in epochs (mean ± std) until peak validation performance. Each cell reports classic cross-entropy as baseline vs. the full LARESA. Lower values indicate faster convergence.

| Model | CIFAR-100 | CUB-200 | TinyImageNet |
|---|---|---|---|
| ResNet50 | **223 ± 22** ǀ 268 ± 11 | 228 ± 19 ǀ **170 ± 9** | 197 ± 24 ǀ **173 ± 14** |
| ResNet101 | **245 ± 20** ǀ 260 ± 15 | 232 ± 18 ǀ **180 ± 11** | 210 ± 20 ǀ **182 ± 12** |
| ResNet152 | **260 ± 25** ǀ 275 ± 16 | 240 ± 20 ǀ **185 ± 13** | 218 ± 21 ǀ **190 ± 13** |
| ConvNeXt-S | 220 ± 18 ǀ **130 ± 9** | 135 ± 12 ǀ **110 ± 7** | 115 ± 10 ǀ **100 ± 6** |
| ConvNeXt-B | 238 ± 24 ǀ **114 ± 7** | 123 ± 15 ǀ **105 ± 3** | **103 ± 4** ǀ 106 ± 6 |
| ConvNeXt-L | 250 ± 22 ǀ **135 ± 10** | 145 ± 13 ǀ **115 ± 8** | 120 ± 12 ǀ **105 ± 7** |
| ViT-Small | 150 ± 15 ǀ **140 ± 12** | 280 ± 20 ǀ **135 ± 10** | 310 ± 18 ǀ **120 ± 8** |
| ViT-Base | 133 ± 18 ǀ **124 ± 10** | 269 ± 16 ǀ **120 ± 9** | 299 ± 12 ǀ **109 ± 7** |
| ViT-Large | 160 ± 17 ǀ **145 ± 11** | 290 ± 22 ǀ **130 ± 11** | 320 ± 20 ǀ **125 ± 9** |

# F    LATENT SPACE VISUALIZATION

ResNet50 on CIFAR-100

(a) $\mathcal{L}_{ce}$          (b) $\mathcal{L}_{ce}, \mathcal{L}_{anchor}, \mathcal{L}_{reg}$          (c) $\mathcal{L}_{ce}, \mathcal{L}_{triplet}, \mathcal{L}_{reg}$          (d) $\mathcal{L}_{total}$ (LARESA)

ConvNeXt-B on CIFAR-100

(e) $\mathcal{L}_{ce}$          (f) $\mathcal{L}_{ce}, \mathcal{L}_{anchor}, \mathcal{L}_{reg}$          (g) $\mathcal{L}_{ce}, \mathcal{L}_{triplet}, \mathcal{L}_{reg}$          (h) $\mathcal{L}_{total}$ (LARESA)

ViT-Large on CIFAR-100

(i) $\mathcal{L}_{ce}$          (j) $\mathcal{L}_{ce}, \mathcal{L}_{anchor}, \mathcal{L}_{reg}$          (k) $\mathcal{L}_{ce}, \mathcal{L}_{triplet}, \mathcal{L}_{reg}$          (l) $\mathcal{L}_{total}$ (LARESA)

| | | | |
|---|---|---|---|
| ■ aquatic mammals | ■ household electrical devices | ■ large natural outdoor scenes | ■ reptiles |
| ■ fish | ■ household furniture | ■ large omnivores and herbivores | ■ small mammals |
| ■ flowers | ■ insects | ■ medium-sized mammals | ■ trees |
| ■ food containers | ■ large carnivores | ■ non-insect invertebrates | ■ vehicles 1 |
| ■ fruit and vegetables | ■ large man-made outdoor things | ■ people | ■ vehicles 2 |

○ Cluster center of fine classes
· Data point

Figure 5: UMAP projections of the embedding space for the experiment on CIFAR-100 with variations in different loss combinations.

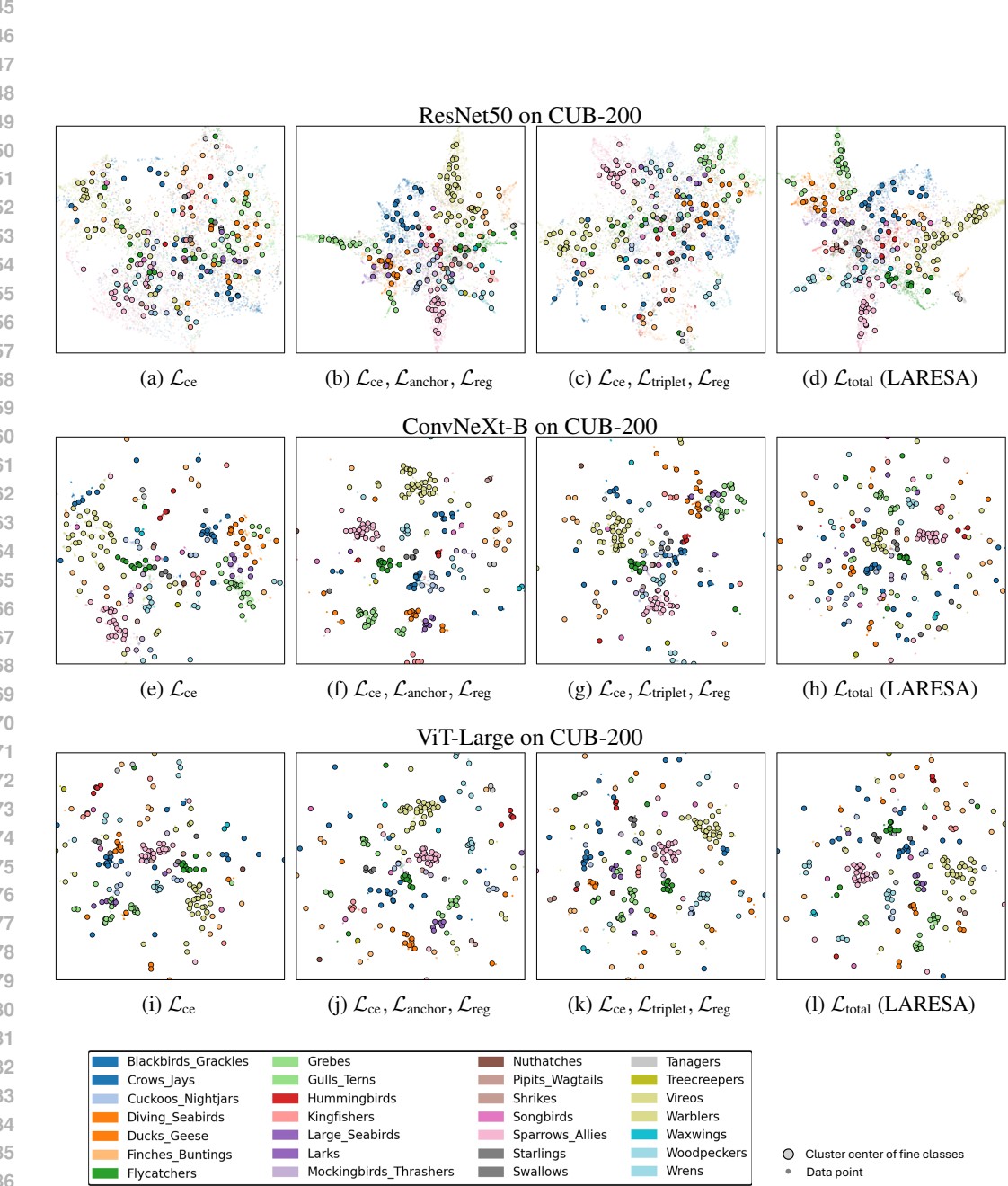

Figure 6: UMAP projections of the embedding space for the experiment on CUB-200 with variations in different loss combinations

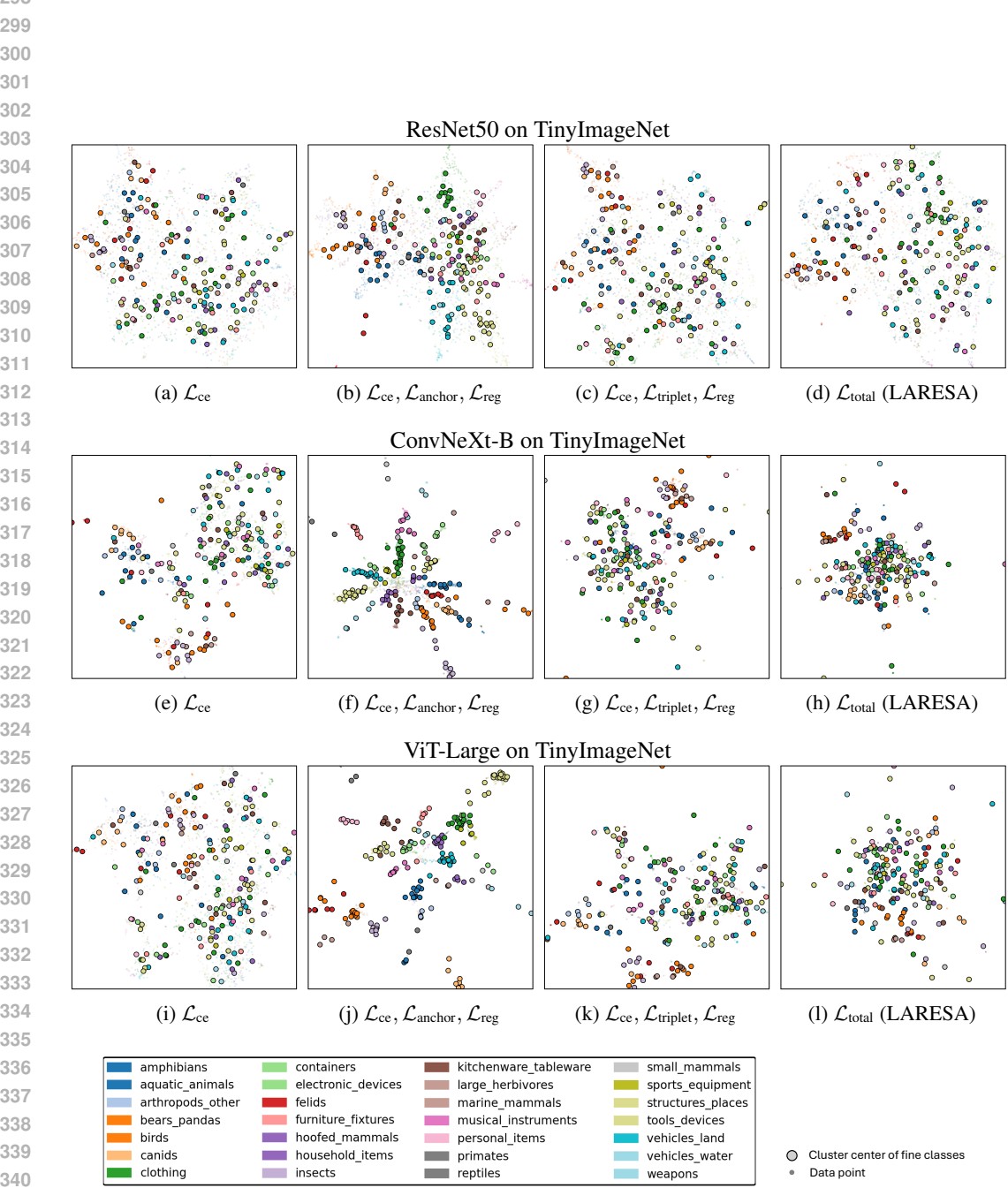

Figure 7: UMAP projections of the embedding space for the experiment on TinyImageNet with variations in different loss combinations

# G   WEIGHTED $\alpha$-LOSS BALANCE

Figure 8: The composition of LARESA loss function with learnable $\alpha$ parameter, tracked for each epoch throughout the training.

## H  WEIGHTED $\beta$-LOSS BALANCE

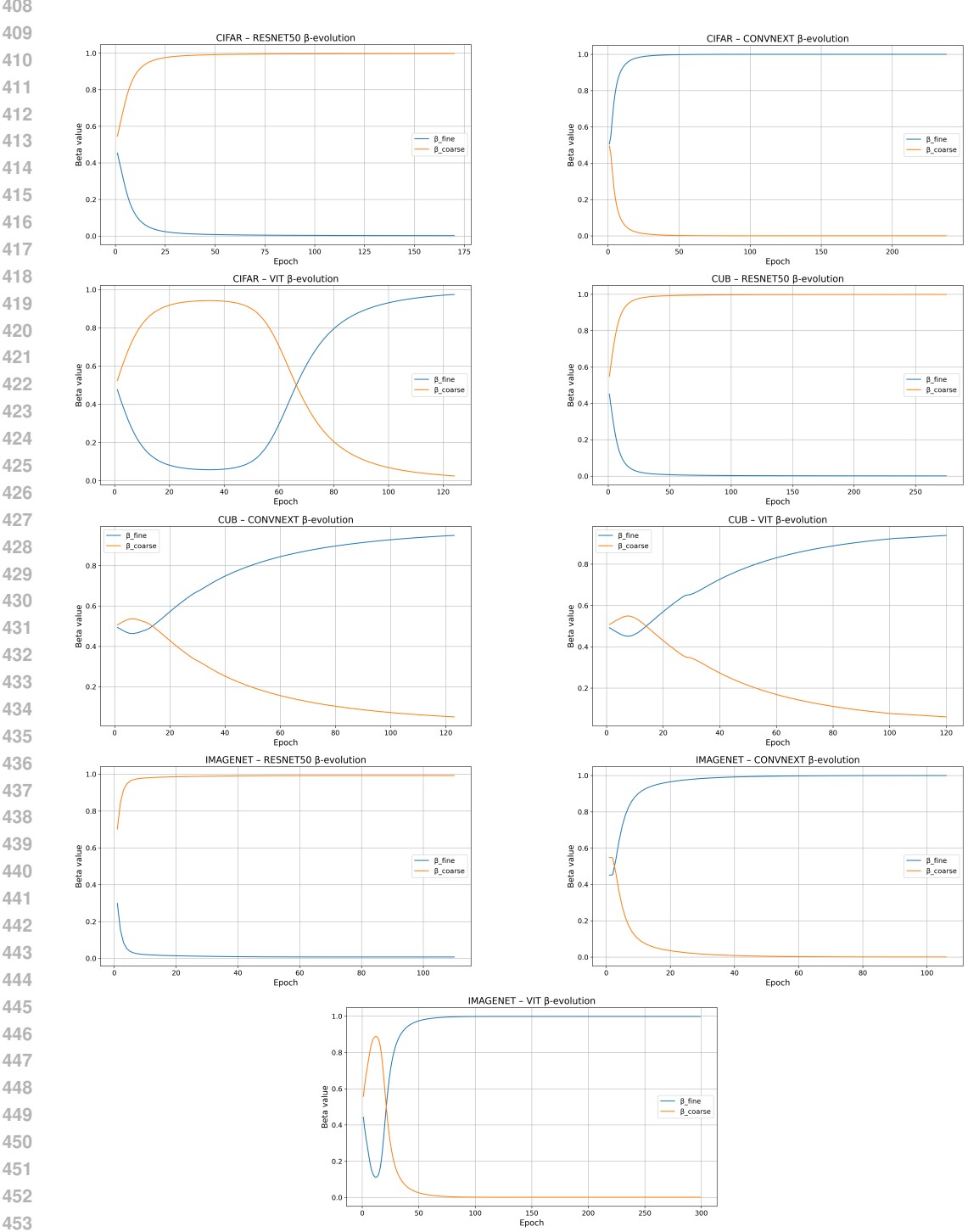

Figure 9: The composition of LARESA $\beta$ weighting function to balance fine and coarse labels, tracked for each epoch throughout the training.

## I  DISCUSSION

**Information-Theoretic Explanation.**   Without any prior knowledge, the model must explore a vast parameter space to find optimal representations for discriminating between classes. This can lead to overfitting and suboptimal generalization performance. Language embeddings capture rich semantic information derived from large text corpora, providing a constraint on the structure of the latent space. This prior knowledge can be viewed as defining a distribution over plausible latent representations, reducing uncertainty and accelerating learning. The mutual information between class labels and their corresponding language embeddings provides an estimate of the expected reduction in entropy gained by incorporating semantic constraints. Incorporating this prior into the training process is analogous to adopting a Bayesian approach: we move from a flat prior distribution over all possible latent spaces to a posterior distribution informed by linguistic knowledge. This bias towards semantically meaningful representations reduces the effective dimensionality of the search space, mitigating overfitting and promoting better generalization performance on unseen data. In essence, our method guides the learning process toward solutions that are not only discriminative but also aligned with external semantic structure.

**Model- and Dataset-Agnostic Design.**   Our process is model-agnostic: the framework applies equally to convolutional and transformer architectures, without requiring changes to the encoder. The only additions are auxiliary loss terms and a learnable weighting mechanism that dynamically adjusts their contribution during training. This enables any standard image classification model to better utilize its latent capacity—especially important in domains with subtle inter-class differences or limited supervision. By relying only on pair-wise similarities in the embedding space, the approach is also independent of latent dimensionality and avoids the alignment problems common in methods that require architectural modifications. LARESA can therefore be applied to any classification task as long as text labels are available.

**Beyond Classification.**   Although demonstrated on image classification, the principle of aligning latent geometry with semantic priors naturally extends further. For regression, continuous labels can be discretized into semantic bins (e.g., "age 0–10," "age 11–20") and mapped into the embedding space. For generative models (VAEs, diffusion), anchoring latent codes to label embeddings could improve controllability and semantic consistency. For multi-task settings, shared semantic anchors across tasks may regularize representation sharing while preserving task-specific nuances. While these directions go beyond the present scope, they highlight that LARESA's design is not tied to classification alone.

**Relation to Prior Latent-Structure Losses.**   Prior work has also sought to impose structure in representation learning, e.g., hierarchical embeddings via CPCC and optimal transport (Zeng et al., 2022; Sinha et al., 2024). These approaches typically require explicitly modeling taxonomies or graph structures, and sometimes architectural modifications. In contrast, LARESA remains a drop-in loss function that automatically adapts its weighting ($\alpha, \beta$) and only requires label text. This positions it as a lightweight alternative: less expressive than full hierarchy embeddings, but far simpler to integrate into standard pipelines.

**When Semantics May Fail.**   Label semantics are not always reliable. In domains with noisy, ambiguous, or highly technical taxonomies (e.g., medical imaging), off-the-shelf embeddings may introduce misleading priors. Our design mitigates this through dynamic weighting: when embeddings are uninformative, the corresponding loss terms quickly down-weight. Moreover, domain-specific embeddings (e.g., BioBERT for biomedical tasks) can substitute SBERT to improve alignment. Still, practitioners should evaluate embedding quality before deployment, and in cases of adversarial or inconsistent labels, plain CE may remain preferable.

**Sustainability and Efficiency.**   A practical benefit of LARESA is faster convergence. Across datasets, less training epochs achieve the same peak validation performance or even higher compared to CE. Although each epoch incurs a small computational overhead from auxiliary losses, the total training time and energy footprint decrease overall. This efficiency angle aligns with recent calls for sustainable ML practice: semantic priors guide optimization into promising regions early, reducing unnecessary computation.

**Comparison with CLIP.** Large-scale multimodal models such as CLIP already encode semantic alignment through pretraining on hundreds of millions of image–text pairs. Applying LARESA to such models yields marginal additional benefit, since semantic structure is already baked in. Instead, LARESA is most attractive in scenarios where CLIP checkpoints are unavailable, unsuitable due to domain shift, or infeasible due to compute or licensing constraints. In such cases, our lightweight class-level alignment offers a complementary alternative: dataset-specific semantic regularization without architectural changes or pretraining cost.

**Limitations and Practitioner Guidance.** We explicitly acknowledge the limitations of this work: (i) we did not test on out-of-distribution benchmarks, though the semantic alignment signal suggests potential benefits; (ii) LARESA's performance depends on embedding quality, which may vary across domains; (iii) auxiliary losses can transiently over-regularize, particularly on ViTs; and (iv) our analysis was limited to in-distribution supervised classification. For practitioners, we recommend monitoring latent metrics (e.g., silhouette score) alongside accuracy. If auxiliary losses cease to improve latent metrics, $\alpha$-weights will typically self-adjust downward; practitioners may also manually freeze them as a safeguard. For hierarchical labels, $\beta$-balancing naturally prioritizes coarse anchors early and fine anchors later, reducing the need for manual scheduling.

**Sensitivity to Embeddings.** Although we used SBERT throughout, the method is embedding-agnostic. In specialized domains, embeddings such as BioBERT (medical) or SciBERT (scientific text) may yield stronger priors. Since losses depend on directional similarity rather than absolute embedding positions, LARESA degrades gracefully if embeddings are noisy, defaulting back toward CE. Future work will explore systematic sensitivity analyses across embedding families.

