# OpenReview forum: "LARESA: Loss-Based Latent Restructuring with Semantic Alignment"
_ICLR.cc/2026/Conference — ICLR 2026 Conference Desk Rejected Submission_

### Official Review · Reviewer_8ZYC · 2025-10-27

**Soundness:** 1
**Presentation:** 2
**Contribution:** 2
**Rating:** 2
**Confidence:** 4

**Summary:**

This paper introduces LARESA, a loss-based regularization framework designed to inject semantic structure into the latent space of standard image classifiers. The method leverages pre-computed text embeddings of class labels (from SBERT) to create semantic priors. It works by adding auxiliary loss terms to the standard cross-entropy objective: a cosine-based anchoring loss to align latent cluster centers with their corresponding text embeddings, a triplet loss to enforce relative semantic distances, and standard geometric regularizers. These components are balanced via a learnable, dynamic weighting mechanism. The authors claim that this architecture-agnostic approach improves classification accuracy and enhances the semantic coherence of the latent space without requiring architectural modifications.

**Strengths:**

- The proposed method is  architecture-agnostic. Its design as a "drop-in" set of auxiliary losses makes it easy to integrate into existing classification training pipelines, at the cost of a more complex final loss.

- Leveraging the inherent semantic information in class labels, which is typically discarded, is a clever and efficient way to introduce more information in the training phase.

- The experiments, within their own setup, demonstrate consistent improvements in accuracy. However, please refer to the weaknesses section for some doubts about the reported baselines performance.

**Weaknesses:**

- The reported performance of the baseline models is a major concern. For instance, the ViT-Large model achieves only  63% top-1 accuracy on CIFAR-100, which is substantially lower than standard benchmarks for a model pre-trained on ImageNet-1k. Similarly, other baseline results appear weak. This raises serious questions about the experimental setup and undermines the significance of the reported gains. Without robust and competitive baselines, the improvements brought by LARESA are difficult to interpret.

- The paper's core motivation, that classifier latent spaces are "sparse, fragmented, and lacking semantic structure", is a strong claim that is not well-supported. It overlooks a significant body of work on emergent alignment [1,2,3,4] and the "Platonic Representation Hypothesis," [5] which suggests that representations from different models can converge to semantically similar structures. The related work section should engage with this literature to properly contextualize the paper's contribution.

- The paper only compares against a simple cross-entropy baseline. However, the standard for learning structured visual representations is now Self-Supervised Learning (SSL) (e.g., Dinov2 [6]). The paper fails to position LARESA relative to these powerful methods. It is unclear whether LARESA provides a meaningful alternative or a complement to SSL, which is a critical omission for a paper focused on improving representation quality.

- Several design choices are not well-motivated. For example, the use of only class names for text embeddings seems limiting; richer descriptions (e.g., from Wikipedia) could provide a stronger semantic signal at a similar one-time computational cost. Furthermore, the necessity of the learnable loss-balancing mechanism is not demonstrated against simpler, non-learnable alternatives like scaling the loss components to yield the same gradient norm at the last layer.

---

[1] Kivva, et al, "Identifiability of deep generative models without auxiliary information," NeuriIPS 2022.

[2] Moschella, et al, "Relative representations enable zero-shot latent space communication," in ICLR, 2023.

[3] Cannistraci, et al, "From Bricks to Bridges: Product of Invariances to Enhance Latent Space Communication," in ICLR, 2024.

[4] Maiorca, et al. Latent Space Translation via Semantic Alignment. In  NeurIPS 2023.

[5] Huh, et al, "The Platonic Representation Hypothesis," 2024.

[6] Oquab, et al, "DINOv2: Learning Robust Visual Features without Supervision.", TMLR 2024

**Questions:**

- Could the authors please clarify the significant discrepancy between the baseline accuracies reported in the manuscript (especially for ViT-Large) and established results in the literature?

- How do the authors reconcile their claim that classifier latent spaces are unstructured with the literature on emergent semantic alignment?

- How does the latent space structure produced by LARESA compare, quantitatively and qualitatively, to that learned by modern SSL methods like DINOv2 [6]? Does applying LARESA to a classifier make its latent space more similar to an SSL-derived space?

- What is the rationale for using only class names instead of richer semantic descriptions (e.g., from Wikipedia), given that text encoding is a one-time offline cost?

- Have the authors evaluated simpler loss balancing schemes to justify the use of the proposed learnable weighting mechanism?

- Could the authors specify which layer's embeddings were used for the latent space analysis? For ViT models, was the [CLS] token used, or an aggregation of patch tokens?

- The reference for TinyImageNet on line 574 appears to link to CIFAR-10. This should be corrected.

- The presented work is closely related to the concept of "relative representations" [2]. The proposed cosine-based triplet loss (in "2.3 Moving Latent Clusters with Relative Semantic Similarity") directly manipulates such relative distances, I strongly suggest citing and discussing this line of work.

- The UMAP visualizations are helpful, but their value would be enhanced by a direct comparison to the latent space of a leading SSL model on the same data. This would help contextualize the quality of the structure LARESA induces.

---

### Official Review · Reviewer_Zvt1 · 2025-10-27

**Soundness:** 1
**Presentation:** 2
**Contribution:** 1
**Rating:** 2
**Confidence:** 4

**Summary:**

The paper introduces LARESA, a framework designed to enhance the class separability and overall representation quality of vision classifiers. This is achieved through two primary contributions: first, by aligning the vision model's semantic space with that of a pre-trained Language Model, and second, by employing a novel, learnable loss function. This composite loss function integrates a standard cross-entropy term with a regularizer, a cosine-based loss, and a triplet-based loss. The authors present empirical results demonstrating that training with the LARESA framework can lead to performance improvements compared to standard training procedures on several datasets.

**Strengths:**

* **S1**: The core idea of combining semantic alignment from an LLM with a dynamically weighted, multi-component loss function is novel.

* **S2**:  The method operates as a modular component of the training pipeline, allowing it to be integrated with a variety of backbone vision architectures, as demonstrated by the authors with both CNN and Transformer models.

* **S3**: The authors have made their source code available.

**Weaknesses:**

While the paper introduces a promising direction, there are several areas that could be strengthened to solidify its contributions:

* **W1. Inconsistent and Underwhelming Results**: The central claim that LARESA improves performance is not consistently supported across all experiments. The optimal combination of the four loss components appears to vary depending on the specific setting, which makes it unclear how to best apply the method in practice. Furthermore, the reported accuracies on standard benchmarks are notably low. For instance, the accuracy for ViT-Large on CIFAR-100 is reported as 63%, whereas fine-tuned ViT-Large models are known to achieve upwards of 90% accuracy (see Table 5 of [1]). Similarly, the results on Tiny ImageNet lag behind established benchmarks for models like ResNet50 and various Vision Transformers [3,4].

* **W2. Clarity of Claims and Presentation**: The assertion in Table 1 that LARESA "yields consistent accuracy gains while also reducing training runtime" could be refined for clarity. As noted, the accuracy gains are not always consistent when using the full loss combination. Regarding the runtime, Appendix E measures this in terms of the number of epochs to reach peak accuracy. A more informative comparison would be the wall-clock time per epoch, which would clarify the computational overhead introduced by the loss calculation.

* **W3. Clarity of Figures and Visualizations**: Some of the figures could be improved for better comprehension. Figure 1, which illustrates the overall framework, is difficult to follow. A clearer diagram would significantly aid in understanding the proposed method. Additionally, the visualization in Figure 3, intended to demonstrate the improved latent space, is not entirely convincing. For example, it is unclear if similarly colored points represent semantically similar classes, and the presence of outliers (e.g., light blue points near violet ones) makes it difficult to understand the improvement of the LARESA space.

* **W4. Limited Experimental Scope**: The empirical evaluation is conducted on a limited set of datasets, and the results are presented without standard deviations, making it difficult to assess the statistical significance of the findings. The choice of Tiny ImageNet over a more standard large-scale benchmark like ImageNet-1k also limits the assessment of the method's scalability. Furthermore, the reliance on a single LLM (all-MINILM-L6-v2) leaves the question of the framework's sensitivity to the choice of the language model unanswered.

* **W5. Practicality and Justification**: The paper does not make a strong case for the practical utility of the proposed method. Given that the framework appears to require training from scratch, the modest performance gains of 1-2% may not justify the associated computational cost.

* **W6. Evaluation of Representation Quality**: The authors claim that their method learns a "good representation." A compelling way to substantiate this claim would be to evaluate the learned features on other downstream tasks. Such experiments would provide more direct evidence of the quality and generalizability of the learned representations.

* **W7. Lack of Comparative Analysis**: The related work section discusses other methods, but the experimental section lacks a direct comparison with these existing approaches. An empirical comparison would be essential to properly situate the performance of LARESA within the current literature.

* **W8: Missing Related Works**: The discussion of related work could be strengthened by incorporating research on representational alignment and emerging similarities within neural networks [4,5,6,7]. Including this context would provide a stronger foundation for the paper's goal of explicitly structuring the latent space.


----
[1] Dosovitskiy, Alexey. "An image is worth 16x16 words: Transformers for image recognition at scale." arXiv preprint arXiv:2010.11929 (2020).

[2] Huynh, Ethan. "Vision transformers in 2022: An update on tiny imagenet." arXiv preprint arXiv:2205.10660 (2022).

[3] Amangeldi, Aidar, et al. "CNN and ViT Efficiency Study on Tiny ImageNet and DermaMNIST Datasets." arXiv preprint arXiv:2505.08259 (2025).

[4] Moschella, Luca, et al. "Relative representations enable zero-shot latent space communication." ICLR (2023)

[5] Maiorca, Valentino, et al. "Latent space translation via semantic alignment." Advances in Neural Information Processing Systems 36 (2023): 55394-55414.

[6] Cannistraci, Irene, et al. "From bricks to bridges: Product of invariances to enhance latent space communication." ICLR (2024).

[7] Huh, Minyoung, et al. "The platonic representation hypothesis." ICML (2024).

**Questions:**

* **Q1**: Are the final results reported on the fine-grained or coarse-grained labels?

* **Q2**: To better understand the contribution of the end-to-end training with LARESA, have the authors considered a simpler baseline? For instance, what are the results if a simple linear classifier is trained for a few epochs on top of the frozen features of a standard pre-trained model?

* **Q3**: Could the authors provide a more detailed analysis of the training time? Specifically, what is the wall-clock time per epoch or per batch when using the full LARESA loss compared to a standard cross-entropy baseline?

* **Q4**:To support the claim of learning a better representation, have the authors performed any experiments using the learned features for other downstream tasks?

* **Q5**: What is the performance of LARESA on a larger and more complex dataset with a greater number of classes, such as ImageNet-1k? This would provide valuable insights into the scalability of the method.

* **Q6**: How does the choice of the LLM influence the final performance? It would be insightful to see results with other, potentially more powerful, LLMs to understand the impact of the semantic guidance on the learned representation.

* **Q7**:  The experiments in the paper show LARESA being applied to models with baseline accuracies that are relatively low for the given datasets. What is the performance impact of LARESA when applied to a model that has already achieved good performance (e.g. VIT-L pretrained)?

---

### Official Review · Reviewer_XrqC · 2025-10-31

**Soundness:** 3
**Presentation:** 3
**Contribution:** 3
**Rating:** 8
**Confidence:** 2

**Summary:**

The paper presents an approach to inject semantic knowledge into the latent space of classifiers to improve accuracy and interpretability. This is achieved with no architecture change, but through a suite of loss terms that are adaptively optimized. The authors present experiments using three image classification transformer and convolutional model families and three datasets, showing a more compact and coherent latent space without sacrificing classification accuracy.

**Strengths:**

- The proposed approach is accessible and easy to apply to existing image classification architectures

- The learnable loss weighting makes the approach more robust and less dependant on manually-set meta parameters

**Weaknesses:**

While the paper is sold as applicable to "any classification task as long as text labels are available", the approach assumes the existence of, or the ability to elicit, a class taxonomy to inform the latent space structuring. This hinders the approach's applicability to tasks such as speech classification, binary tasks like spam detection, flat tasks like sentiment analysis or face recognition, or multi-label, multi-class tasks.

**Questions:**

- The main thesis of the work is that the taxonomy inherent in the textual class labels is useful when enforced on the richer latent space of a downstream task (e.g. image classification) to improve generalisation. To what extent does that limit transfer? E.g. would the image features learnable for some task be of any use to another classification task?

- The embedding model of the trained architecture needs to have the same dimensionality as SBERT model. Is not that an architectural change? How lossy is this transformation to the expressivity of the original model's latent space?

- The paper claims that the internal semantic coherence is useful beyond the classification accuracy gains and/or efficiency. Could this be demonstrated, e.g. by out of distribution benchmarks?

---

### Official Review · Reviewer_w4Sn · 2025-11-01

**Soundness:** 2
**Presentation:** 3
**Contribution:** 2
**Rating:** 4
**Confidence:** 4

**Summary:**

This paper proposes a method called LARESA to incorporate semantic priors into deep classification models based on class label texts by adding auxiliary loss terms into loss function. The semantic priors are language embeddings of class label texts obtained through the SBERT language model during preprocessing. By introducing three additional loss terms as regularization for the semantic alignment, the paper use learnable logits to balance different loss terms and obtained a new loss function for the deep classification models. The experimental results show that with the new loss function, the Top-1 accuracy of three family of models improves over three datasets.

**Strengths:**

(1) The paper is well written and well organized. It uses clear mathematical formalization to illustrate the design of loss function. The motivation of introducing such design is also explanation in many cases, which make the paper easy to follow.

(2) The LARESA method proposed in the paper is formalized rigorously and technically sound. A cosine loss term is defined by using the language embeddings of class label text as anchors for positioning of latent representations and a marginal loss is defined to separate latent representations for different class by pushing them away.

(3) The paper conducts ablation study and addition experiments to verify the learning result of the proposed model is consistent with the design objective, which strengthens the paper quality.

**Weaknesses:**

(1) The experiments of the paper are only performed on small dataset. The semantic relations in these datasets are relatively simple. It would be interesting to see whether this method can be also applied to larger datasets with more complex semantic relations.

(2) According to the Appendix F, in many models and datasets, the visualizations of latent space are not shown a clear and consistent semantic structures in many cases. However, the Figure 3(b) displays a very clear structure. It would be better to have a more deep analysis to explain this observation.

(3) The semantic correlation of latent space experiment in section 4.4 is not well designed and analyzed. It is clear that without alignment training, the latent representations will not have correlation with the embeddings generated by the language model. It is also obvious that the model trained with loss term to enforce the alignment between language embeddings and the latent representations should have higher Spearman and Pearson correlations. This is not evidence that the models are **improved**. It is only an indication that the result models obtain the semantic alignment as your design.

**Questions:**

(1) Some mathematical notations used in the paper is not defined or explained before use. For example, $L_{ce}$ in section 2.5 and $L_{variance}$ in the Appendix A.

(2) In section 2.4, how these two regularizations are used in the proposed loss is not clearly stated.

---

### Note · Program_Chairs · 2026-01-17
**Submission Desk Rejected by Program Chairs**

The following references in this submission do not refer to real documents and/or have major errors in bibliographic information:

 Hao Liu, Rowan Zellers, Ari Holtzman, Ali Farhadi, Yejin Choi, and Noah A Smith. Learning better visual representations with self-supervised alignment of speech and vision. In NeurIPS, 2021.
Yinan Huang, Jiaming Zhang, and Shizhao Bian. Exploring generalization limits of cross-entropy and mse losses in neural networks. In International Conference on Learning Representations (ICLR), 2022.